# Context-Aware Learning Curve Extrapolation with Prior-Data Fitted Networks

Cheng Yan [1]   Steven Adriaensen [2]   Tom Viering [1]

## Abstract

Extrapolating learning curves of machine learning algorithms can help estimate future performance at larger training set sizes and inform data collection budgets, yet remains challenging on heterogeneous tabular benchmarks where curve shapes vary across learners and tasks. We revisit real-data training for Learning Curve Prior-Data Fitted Networks and find that direct training often overfits. We first introduce Real-LC-PFN++, a stabilized real-data baseline that combines early stopping and a content-free learnable dummy token (also called register token). Building on this, we propose CTX-LC-PFN, a context-aware extrapolator that conditions its predictive distribution on auxiliary context, specifically, the learner identity and observed training performance. In the largest learning curve benchmark, LCDB 1.1, our methods improve uncertainty calibration and extrapolation accuracy across observation regimes. Code is available at https://github.com/learning-curve-research/CTX-LC-PFN.

## 1. Introduction

Learning curves are a fundamental concept in machine learning, characterizing how an algorithm's performance evolves as a function of experience. In this context, it is important to distinguish between two common notions of experience. Epoch-wise learning curves, also referred to as training curves, measure performance or optimization loss as a function of training iterations or epochs. In contrast, sample-wise learning curves describe the relationship between a model's performance and the training set size. These curves (hereafter referred to as *learning curves*) serve as a practical tool for data-centric decision making (Viering & Loog,

2022; Mohr & van Rijn, 2024), such as data requirement estimation (Mahmood et al., 2022a).

Modeling learning curves, especially extrapolating performance from smaller training datasets to larger ones, is often approached via parametric families (e.g., power laws or exponentials) (Gu et al., 2001; Figueroa et al., 2012; Kaplan et al., 2020; Mohr et al., 2022; Alabdulmohsin et al., 2022). These require strong functional-form assumptions that are brittle when curve shapes vary substantially across learners and tasks, as is the case for tabular data (Viering & Loog, 2022; Yan et al., 2025). More recently, Adriaensen et al. (2023) propose Learning Curve Prior-Data Fitted Networks (LC-PFN), training a transformer in an in-context learning paradigm (Brown et al., 2020; Müller et al., 2022) to approximate Bayesian inference over learning curves, but adopt the carefully hand-designed synthetic prior from Domhan et al. (2015). In the sample-wise learning curve setting, Viering et al. (2024) introduce Real-LC-PFN and show that directly training on real learning curves can outperform LC-PFN variants based on hand-designed synthetic priors.

Our work finds that training Real-LC-PFN directly on real learning curve data often overfits without carefully tuning hyperparameters (one example shown in Figure 1a). This suggests that, while real-data training is a promising alternative to hand-designed priors, its effectiveness can be further improved, motivating the following question:

> *How can LC-PFN be effectively trained directly on real learning curve data, rather than relying on carefully hand-designed synthetic priors?*

First, we revisit Real-LC-PFN under a validation protocol and show that late-stage overfitting is a major failure mode. Here, we explore effective early stopping and find that adding a content-free learnable dummy token, inspired by register tokens (Darcet et al., 2024), further stabilizes training and improves downstream predictive performance. This leads to Real-LC-PFN++, a solid baseline for pretraining in-context learning curve extrapolation on real data.

In addition, while pretraining on a heterogeneous mix of datasets and learners improves uncertainty estimates, it often (correctly) results in high uncertainty when extrapolating from a partial curve alone, reflecting a high-entropy posterior under a broad empirical prior. To reduce this un-

[1]Pattern Recognition & Bioinformatics Group, Delft University of Technology, Delft, the Netherlands [2]Machine Learning Lab, University of Freiburg, Freiburg, Germany. Correspondence to: Cheng Yan <c.yan-1@tudelft.nl>, Tom Viering <t.j.viering@tudelft.nl>.

*Proceedings of the $2^{nd}$ ICML Workshop on Foundation Models for Structured Data*, Seoul, South Korea. 2026. Copyright 2026 by the author(s).

**(a)** 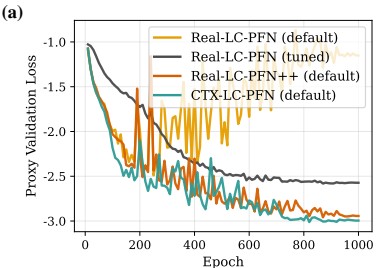
**(b)** 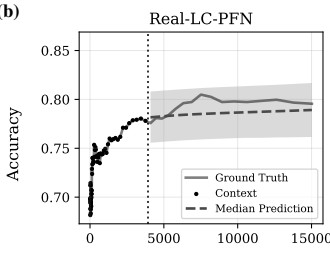 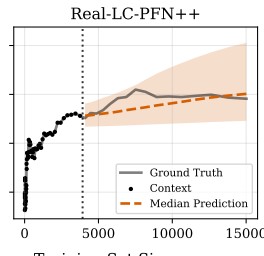 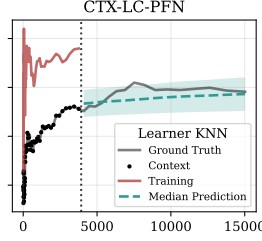

*Figure 1.* **Inference example and training stability.** (a) Proxy validation loss over training epochs, computed as negative log-likelihood (NLL, ↓) on unseen validation datasets and averaged over a range of cutoff values (Appendix A.1). (b) Example extrapolation from a given partial learning curve, where the model predicts future generalization accuracy.

certainty, we condition predictions additionally on two relevant and readily available information sources: (i) **learner identity**, since the shapes of curves are often algorithm-specific (Yan et al., 2025); and (ii) **training-set performance**, which can provide a useful signal about task difficulty (Swayamdipta et al., 2020). To this end, we propose CTX-LC-PFN, a Context-Aware Learning Curve Prior-Data Fitted Network that conditions the predictive distribution on auxiliary context; see Figure 1b. On the largest learning curves benchmark, LCDB 1.1 (Yan et al., 2025), CTX-LC-PFN improves both extrapolation accuracy and uncertainty calibration across observation regimes.

## 2. Preliminaries & Related Work

**Learning Curve Modeling.** Sample-wise learning curves characterize the relationship between training set size and model performance. A common approach is to fit parametric functions, such as power laws or exponentials, to support performance prediction (Gu et al., 2001; Figueroa et al., 2012). While scaling laws provide a robust parametric framework for deep learning (Kaplan et al., 2020; Alabdulmohsin et al., 2022), they often struggle with tabular data, where curves can be non-monotone or exhibit diverse shapes (Viering & Loog, 2022; Yan et al., 2025; Cazacu et al., 2025). Indeed, in these settings, parametric models often underperform the simple but strong heuristic of last-value extrapolation (Mohr et al., 2022; Kielhöfer et al., 2024). In light of these limitations, Bayesian approaches (Klein et al., 2017b;a; Garnelo et al., 2018a;b; Gordon et al., 2020; Müller et al., 2022) provide more flexible, data-driven extrapolators. Inspired by Müller et al. (2022), LC-PFN (Adriaensen et al., 2023) reframes the extrapolation task in the in-context learning framework by pretraining on synthetic learning curves generated from a pre-designed prior (Domhan et al., 2015) that avoids committing to a single restrictive parametric form. LC-PFN produces accurate predictions with well-calibrated uncertainties in a single forward pass. This paradigm has been extended to sample-wise curves by incorporating real data (Viering et al., 2024) and to scaling curves (Lee et al., 2025).

Building on prior work, we focus on the robustness of direct training on real learning curves and study how conditioning on auxiliary context affects the predictive distribution over future curve values. Our content-free learnable dummy token is closely related to register tokens and attention sink, which introduce auxiliary positions for internal computation or routing rather than sample-specific information (Darcet et al., 2024; Jiang et al., 2025; Xiao et al., 2024). To the best of our knowledge, this is the first application of such content-free tokens to PFNs. More broadly, related work has explored real-data continued pretraining of PFNs in tabular data (Garg et al., 2025; Grinsztajn et al., 2025).

Conditioning PFNs on prior information about the optimum has previously been explored in black-box Bayesian optimization (Müller et al., 2023; Viering et al., 2025) and in freeze-thaw hyperparameter optimization (Rakotoarison et al., 2024). Outside the PFNs framework, auxiliary-context conditioning has also been explored for learning-curve extrapolation and forecasting, including conditioning on hyperparameter configurations (Klein et al., 2017b; Jawed et al., 2021), dataset meta-features (Jawed et al., 2021), neural architecture descriptors (Ding et al., 2025), and task difficulty estimates (Li & Zhao, 2026). These approaches differ from ours either in setting (e.g., epoch-wise or architecture-conditioned curves) or in paradigm (single-task training rather than in-context inference).

**Performance Prediction & Data Collection.** Learning curve extrapolation can help both performance prediction and data collection planning across domains, including tabular data (Figueroa et al., 2012), language (Kolachina et al., 2012; Alabdulmohsin et al., 2022), and computer vision (Mahmood et al., 2022a; Jain et al., 2023). Although most extrapolators are evaluated in performance space, downstream data collection decisions are more budget-sensitive: the same prediction error can have different consequences at different training set sizes. Accordingly, Mahmood et al. (2022b; 2025) formulates data collection as a cost-aware sequential optimization problem and introduces the Learn-Optimize-Collect (LOC) framework. However, in contrast to our learning curve extrapolation approach, LOC is itera-

tive and requires additional cost definitions.

## 3. Methods

**Problem setup.** We consider sample-wise learning curve extrapolation. Given a sequence of training set sizes $x_{1:N} = (x_1, \ldots, x_N)$ and the corresponding generalization performances $y_{1:N}$, we observe only a partial curve $\hat{\mathbf{C}}_{1:n} = \{(x_i, y_i)\}_{i=1}^n$ with a cutoff $n < N$, and aim to predict the future performance values $y_{n+1:N}$ at larger training set sizes. Real-LC-PFN trains the transformer-based probabilistic model LC-PFN (Adriaensen et al., 2023) directly on real-world sample-wise learning curves with data augmentation (Viering et al., 2024). For each future query input $x_i, i = n + 1, \ldots, N$, this model parameterizes a predictive distribution $q_\theta(y_i \mid \hat{\mathbf{C}}_{1:n}, x_i)$, and is trained by minimizing the negative log-likelihood (NLL) over the extrapolation portion. Without additional context, the model approximates a marginal distribution over heterogeneous learners and datasets, so similar observed prefixes from different learners or tasks can remain ambiguous.

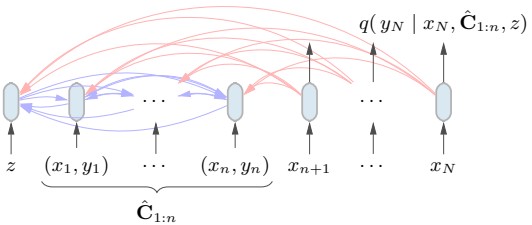

$$q(y_N \mid x_N, \hat{\mathbf{C}}_{1:n}, z)$$

*Figure 2.* **Overview of the CTX-LC-PFN architecture.** Context tokens $z$ are prepended to the observed learning curve, where $z$ can be learner identity $z_L$, last observed training-set performance $z_T$, or both as separate tokens; the PFN then predicts generalization performance at larger training set sizes in one forward pass. Blue curved arrows denote attention within the conditioning set $\{z\} \cup \hat{\mathbf{C}}_{1:n}$, while red curved arrows denote predictions at future query inputs reading this set independently, with no query–query attention; upward arrows are prediction heads.

**Real-LC-PFN++.** We introduce Real-LC-PFN++ as a stabilized LC-PFN variant for direct training on real learning curves. To mitigate the overfitting observed in Figure 1a, Real-LC-PFN++ uses early stopping based on the proxy validation loss (see Appendix A.1) and prepends a content-free learnable dummy token. Specifically, it follows the same schematic as Figure 2, but instantiates $z$ as a single randomly initialized dummy token.

**CTX-LC-PFN.** In contrast, CTX-LC-PFN uses the same architecture but replaces the content-free token with informative auxiliary context. Specifically, it conditions the predictive distribution at each future query input on compact context tokens $z$:

$$q_\theta(y_i \mid \hat{\mathbf{C}}_{1:n}, x_i, z), \quad i = n + 1, \ldots, N,$$

as illustrated in Figure 2. We instantiate $z$ with learner identity ($z_L$) and the last observed training-set performance ($z_T$), which capture algorithm-specific curve behavior and task difficulty. The learner identity specifies which learning algorithm generated the curve, and is encoded by a one-hot learner indicator followed by a learned embedding lookup, yielding a single token $z_L$. When both context sources are used, $z_L$ and $z_T$ are prepended as separate tokens. We evaluated a variant that keeps an additional content-free dummy token together with the context tokens, but it did not further improve performance; see Appendix B.1. We also tried conditioning on each context $z_L$ or $z_T$ individually and chose to use both; see Appendix B.2. For interpretability, Section 5 analyzes the geometry of the learned learner-token.

## 4. Experimental Setup

**Dataset.** We evaluate on LCDB 1.1 Clean, a curated subset of Learning Curves Database 1.1 (LCDB 1.1), the largest learning curve benchmark spanning diverse tabular classification tasks and algorithms (Yan et al., 2025); see Appendix D. For each learner–dataset pair, LCDB 1.1 provides training, validation, and test performance curves under multiple random splits; we use validation set accuracy as the generalization performance $y$ to extrapolate. We use dataset-level splits for out-of-distribution evaluation, so validation and test curves come from unseen datasets while learner identities remain known at test time.

**Model and Training Configuration.** We compare the pretrained LC-PFN, Real-LC-PFN (tuned but no early stopping), Real-LC-PFN++, and CTX-LC-PFN by minimizing extrapolation NLL. In addition, we include two parametric baselines: *last1*, which predicts all future values using the last observed value, and *exp4*, a four-parameter exponential model $y = c - \exp(-ax^d + b)$. For probabilistic prediction, we use training set to learn a frequency-based binned distribution. More details see Appendix A.

**Evaluation Metrics.** We evaluate probabilistic quality with negative log-likelihood (NLL), mean squared calibration error (MSCE) (Kuleshov et al., 2018), and quantile loss (Koenker & Hallock, 2001), and point prediction with mean squared error (MSE) and root mean squared log-error (RMSLE). To assess data requirement estimation, we define data estimation error, $\mathcal{E}_{\text{data}} = \frac{1}{M} \sum_{i=1}^M |x_i(y_i - \hat{y}_i)|$, where $M$ is the number of evaluated extrapolation points and $\hat{y}_i$ denotes the predicted performance. See Appendix E for metric details and motivation.

## 5. Results

CTX-LC-PFN improves predictive likelihood and uncertainty calibration across observation regimes. It outperforms all the other models on NLL, MSCE, and quantile

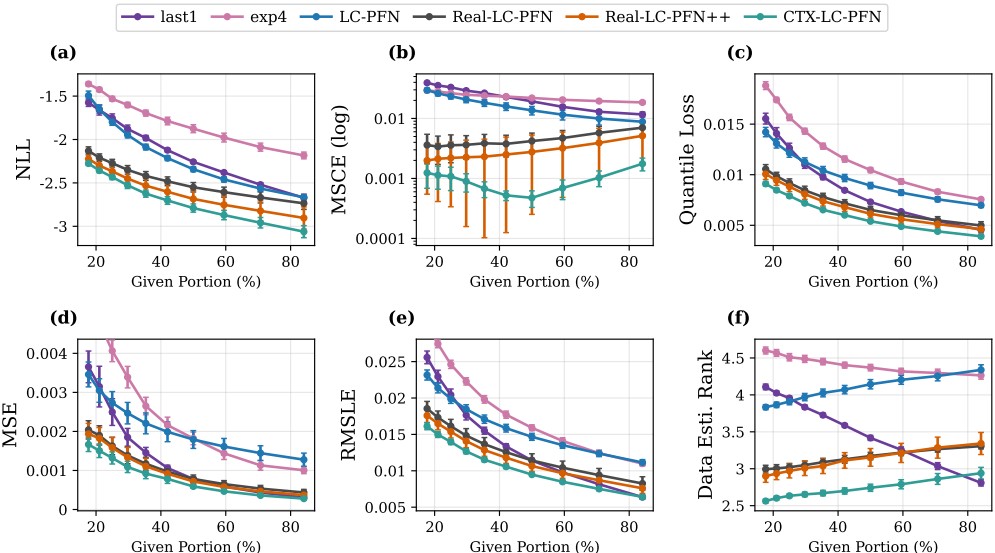

*Figure 3.* Benchmarking predictive uncertainty, performance prediction, and data estimation error. Metric details are given in Appendix E. Error bars indicate the standard error over 5 random splits of the data. (a) Negative log-likelihood (NLL, ↓). (b) Mean squared calibration error (MSCE; (Kuleshov et al., 2018), ↓). (c) Quantile loss (Koenker & Hallock, 2001) (or pinball loss, ↓). (d) Mean squared error (MSE, ↓). (e) Root mean squared log-error (RMSLE, ↓). (f) Rank on data estimation error (↓).

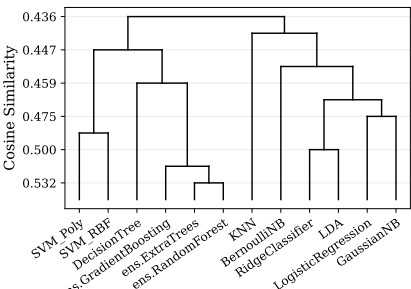

*Figure 4.* Average-linkage hierarchical clustering based on mean cosine similarity across random seeds.

loss, while Real-LC-PFN++ serves as the second strongest baseline (Figure 3a–3c, NLL rank statistical significance see Appendix F). Moreover, as the observed portion increases, all methods generally improve in NLL and quantile loss.

Beyond distributional metrics, CTX-LC-PFN also improves performance prediction and data requirement estimation. Figures 3d and 3e show its more modest but consistent MSE/RMSLE advantage over the other PFN-based methods. For MSE, the gain is larger in the low-observation regime, where auxiliary context helps with limited prefixes; as more of the curve is observed, the gap narrows. For RMSLE, however, the gap appears more uniform. Figure 3f reports ranks of data estimation error as a proxy for cost estimation when mis-specify the data requirement. CTX-LC-PFN achieves consistently better ranks than the other PFN-based methods. In the high-observation regime, however, *last1* performs best overall, indicating that once most of the curve

has already been observed, a simple local extrapolation from the latest point becomes a very strong baseline, consistent with prior work (Mohr et al., 2022; Kielhöfer et al., 2024).

Lastly, we provide an interpretability view of the model's use of learner identity. Figure 4 visualizes the learned learner-token space using average-linkage clustering based on cosine similarity. This suggests that the learned token space encodes behaviorally meaningful variation across learners, consistent with the learner diversity documented in LCDB 1.1 (Yan et al., 2025); see Appendix C.

## 6. Conclusion & Future Work

To conclude, we revisited direct real-data training for LC-PFN in sample-wise learning curve extrapolation. We found that training on heterogeneous real learning curves can over-fit on unseen datasets, and introduced Real-LC-PFN++ as a stronger real-data baseline. Building on this, we proposed CTX-LC-PFN, an in-context extrapolator that conditions its predictive distribution on auxiliary context, including learner identity and observed training-set performance. On LCDB 1.1 Clean, CTX-LC-PFN improves extrapolation quality and uncertainty calibration across observation regimes, suggesting that context can help disambiguate future extrapolation of partially observed learning curves.

Future work could extend this setting to multiple observed curves (Rakotoarison et al., 2024; Ruhkopf et al., 2023) and evaluate generalization in broader benchmarks (Turan et al., 2026) and more downstream tasks, such as data acquisition with Learn-Optimize-Collect (Mahmood et al., 2025).

## Acknowledgements

We are grateful to Arber Zela for helpful input on an earlier draft of this work. We also thank the anonymous reviewers for their constructive comments and suggestions.

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

# A. Model Hyperparameters and Baseline Calibration

### A.1. Backbone Setup and Training with Proxy Validation Loss

When we train LC-PFN directly on real learning curves, we need to reserve held-out datasets to monitor generalization during training. Specifically, for each random seed, the LCDB 1.1 Clean learning curve datasets are split into training, validation, and test datasets as described in Appendix D. The model is trained only on curves from the training datasets, while we monitor a proxy validation loss during training and use it only for light hyperparameter tuning and early stopping.

Since LC-PFN is an in-context model, evaluating the validation NLL for every possible cutoff at every checkpoint would be expensive. We therefore use a fixed set of cutoffs as a computationally efficient proxy. Since the learning curves have different lengths, and the shortest curves contain 40 points, we average the validation NLL over extrapolation points for a fixed set of cutoffs, namely 4, 8, 12, and 16 (equivalent to being given approximately 59%, 42%, 30%, and 21% of the entire curves). This differs from training, where cutoffs are sampled randomly, and makes the validation loss more stable and comparable across checkpoints. The resulting loss is first averaged over all eligible validation prediction points for each cutoff and then averaged over the selected cutoffs, yielding a single scalar proxy validation loss for each checkpoint. We evaluate this proxy loss every 10 training epochs and the early-stopped model used in our experiments is the checkpoint with the lowest proxy validation loss.

To check the suitable scale of LC-PFN models, we train LC-PFN backbones at three model scales (Figure 5). The small backbone (approximately 0.69M parameters) has 3 layers, 1 attention head, an embedding size of 128, and a feed-forward hidden size of 256. The medium backbone (approximately 3.81M parameters) has 6 layers, 2 attention heads, an embedding size of 256, and a feed-forward hidden size of 512. The large backbone (approximately 26.79M parameters) has 12 layers, 4 attention heads, an embedding size of 512, and a feed-forward hidden size of 1024. All of them use dropout of 0.2. Instead of predicting a single value, all models output a binned predictive distribution over $y$ with 1000 bins. For each data split, we fit a 1000-bin equal-frequency discretization of $y$ using only finite generalization accuracy values from the training datasets, save the resulting criterion with the checkpoint, and reuse it unchanged for validation and held-out test evaluation so that no validation or test targets are used to define the bin edges.

Training is run for 1000 epochs with 100 steps per epoch and a batch size of 100, using curve augmentation (Viering et al., 2024). Curve augmentation is applied before the model tokens are constructed: the implementation samples 10 points uniformly in $[0, 1]$, considers all pairs as candidate target ranges, and applies the affine map of the curve values whose transformed curve has the smallest squared deviation from the original. For variants using training-set performance context, the same affine transformation is applied jointly to the generalization accuracy and training accuracy curves so their scale relationship is preserved. We optimize with Adam (Kingma, 2015) with a learning rate of $10^{-4}$ and gradient clipping at 1.0. We use a cosine annealing learning rate schedule (Loshchilov & Hutter, 2017) with a linear warmup over the first 25% of training.

Monitoring the proxy validation loss reveals that directly training LC-PFN on heterogeneous real learning curves with the default Real-LC-PFN training configuration (Viering et al., 2024; Adriaensen et al., 2023) can easily overfit. Across model scales, the training NLL can keep improving while the proxy validation loss and extrapolation quality on unseen datasets degrade. Figure 5 compares Real-LC-PFN models trained at the model scales described above with the default settings, showing that this overfitting behavior is not specific to a single backbone size. Unless stated otherwise, we use the medium-size backbone.

### A.2. Real-LC-PFN++ Configuration

We keep the model medium-size backbone and focuses on the learning rate, dummy token, and checkpoint selection. We tune the peak learning rate over $\{10^{-3}, 10^{-4}, 10^{-5}\}$ under the same warmup and cosine schedule. Based on this sweep, the tuned Real-LC-PFN uses a peak learning rate of $10^{-5}$ and the final checkpoint, while Real-LC-PFN++ and CTX-LC-PFN use a peak learning rate of $10^{-4}$ and early stopping based on the proxy validation loss in Appendix A.1. Figure 1a compares these trajectories under the proxy validation loss.

Figure 6 examines the two key design choices in Real-LC-PFN++: learning rate and early stopping. Panel (a) shows proxy validation loss curves for Real-LC-PFN and Real-LC-PFN (Dummy) trained at lr$\in \{10^{-3}, 10^{-4}, 10^{-5}\}$. Across both model families, lr$= 10^{-4}$ achieves the best proxy validation loss: it converges faster than lr$= 10^{-5}$ while avoiding the instability seen at lr$= 10^{-3}$. Adding a content-free learnable dummy token consistently lowers the proxy validation loss at each learning rate, confirming its role as a stabilizing component independent of the learning rate choice. Panel (b) shows the

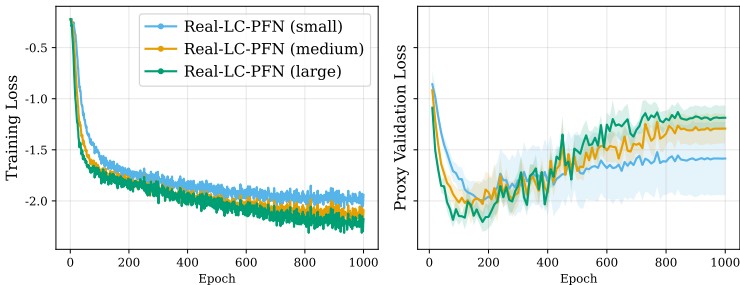

*Figure 5.* **Real-LC-PFN overfitting under the default training configuration across model scales.** Training with the default Real-LC-PFN settings can continue to reduce training loss while the proxy validation loss on unseen validation datasets increases after its best checkpoint. Training (left) and proxy validation loss (right) curves averaged over 5 random seeds (standard error as shading).

downstream impact on held-out test NLL. Without early stopping, Real-LC-PFN (Dummy, lr$= 10^{-4}$, full training) is worse than its early-stopped counterpart, showing that late-stage overfitting persists even with the dummy token. Real-LC-PFN++ (Dummy, lr$= 10^{-4}$, early stopping) outperforms Real-LC-PFN (lr$= 10^{-5}$, full training) across all observed portions, demonstrating that combining the dummy token with a moderate learning rate and early stopping is more effective than relying on a small learning rate alone to prevent overfitting.

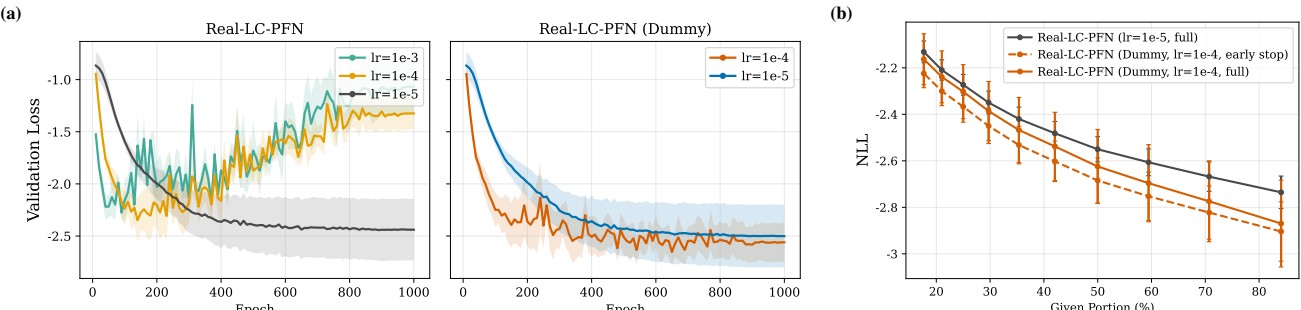

*Figure 6.* **Real-LC-PFN++ ablation: learning rate and early stopping.** (a) Proxy validation loss (NLL, $\downarrow$) over training epochs (mean $\pm$ SE across 5 seeds) for Real-LC-PFN and Real-LC-PFN (Dummy) at three learning rates. (b) Held-out test NLL across observed portions for three selected checkpoints: Real-LC-PFN (lr$= 10^{-5}$, full training), Real-LC-PFN++ (Dummy, lr$= 10^{-4}$, early stopping), and Real-LC-PFN (Dummy, lr$= 10^{-4}$, full training).

### A.3. Deterministic Baseline Calibration

The mean prediction of *last1* is the last observed value, repeated for all future values. For *exp4*, we fit the four parameters separately for each prefix by least-squares curve fitting with the Levenberg–Marquardt algorithm (Moré, 2006). We repeat the optimization from random initializations, discard fits with NaN or infinite extrapolations, and keep the fit with the smallest prefix mean squared error.

For probabilistic prediction, we convert each point prediction baseline into a binned predictive distribution. For each baseline, held-out suffix cutoff $K$, and tail step $j \in \{1, \ldots, K\}$, we estimate a Gaussian residual scale $\sigma_{K,j}$ on the training datasets from the residuals between the baseline mean prediction and the observed future generalization value. For held-out test evaluation, the baseline prediction is used as the Gaussian mean and the corresponding training-estimated $\sigma_{K,j}$ is used as its standard deviation. We then truncate this Gaussian to the valid generalization-accuracy range, implemented as $[10^{-8}, 1]$ for numerical stability, renormalize it, and integrate its probability mass over bins fitted from the training datasets with the same frequency-based construction as the PFN output distribution. This yields normalized bin masses that are evaluated with the same binned-density likelihood as the PFN models. We do not refit either the uncertainty scale, the binning, or the deterministic model parameters using validation or held-out test future generalization values.

# B. Ablation study

### B.1. CTX-LC-PFN Dummy Token Ablation

We evaluated a CTX-LC-PFN variant that keeps the content-free dummy token in addition to context token $z_L$ (learner identity) and $z_T$ (last observed training performance), using the same training and early stopping protocol as the default CTX-LC-PFN. Figure 7 shows that the extra content-free token did not improve aggregate predictive metrics over the default replacement design. We therefore use the formulation in which the Real-LC-PFN++ dummy token is replaced by informative auxiliary context tokens.

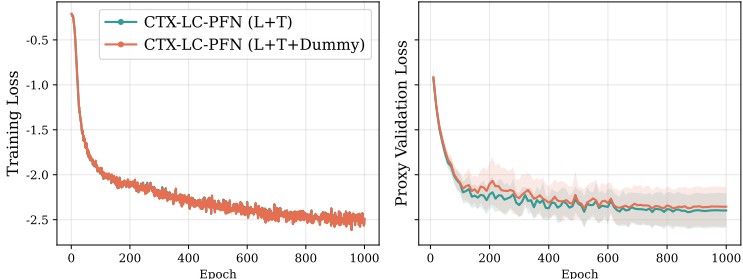

*Figure 7.* **CTX-LC-PFN training and proxy validation loss: (L+T) vs.(L+T+Dummy).** Training (left) and proxy validation loss (right) curves averaged over 5 random seeds (standard error as shading). Adding a content-free dummy token alongside $z_L$ and $z_T$ does not yield a consistent reduction in validation loss.

### B.2. CTX-LC-PFN Conditioned Context Ablation

*Table 1.* Learner-Dataset-level NLL win rate of CTX-LC-PFN against each opponent across observed-curve portions, sorted in ascending order. For each cutoff and seed, one comparison unit is a dataset–learner pair; CTX-LC-PFN wins the pair if it achieves lower NLL on a majority of the corresponding extrapolation points. Entries report mean±std over 5 seeds, and values above 50% indicate that CTX-LC-PFN wins more dataset–learner pairs than the opponent.

| Opponent \ Given | 18% | 21% | 25% | 30% | 35% | 42% | 50% | 59% | 71% | 84% |
|---|---|---|---|---|---|---|---|---|---|---|
| vs. Real-LC-PFN | $72.1 \pm 4.9\%$ | $72.3 \pm 5.8\%$ | $72.7 \pm 5.3\%$ | $75.0 \pm 5.9\%$ | $76.1 \pm 6.0\%$ | $78.8 \pm 6.0\%$ | $81.6 \pm 6.0\%$ | $83.5 \pm 5.3\%$ | $86.9 \pm 4.4\%$ | $86.5 \pm 5.0\%$ |
| vs. Real-LC-PFN++ | $61.8 \pm 5.4\%$ | $63.8 \pm 6.0\%$ | $63.2 \pm 5.8\%$ | $63.4 \pm 6.6\%$ | $65.6 \pm 7.5\%$ | $68.0 \pm 8.8\%$ | $67.7 \pm 12.2\%$ | $68.8 \pm 13.5\%$ | $70.1 \pm 13.8\%$ | $68.6 \pm 16.4\%$ |
| vs. CTX-LC-PFN (L) | $51.0 \pm 10.6\%$ | $52.8 \pm 12.3\%$ | $53.1 \pm 11.5\%$ | $52.7 \pm 12.4\%$ | $54.5 \pm 13.6\%$ | $53.0 \pm 13.8\%$ | $53.3 \pm 14.9\%$ | $53.0 \pm 16.7\%$ | $52.3 \pm 16.7\%$ | $49.6 \pm 18.0\%$ |
| vs. CTX-LC-PFN (T) | $57.9 \pm 8.6\%$ | $58.8 \pm 9.4\%$ | $58.4 \pm 10.3\%$ | $59.9 \pm 11.3\%$ | $58.8 \pm 15.0\%$ | $58.7 \pm 15.6\%$ | $57.7 \pm 17.4\%$ | $58.0 \pm 19.0\%$ | $58.9 \pm 20.4\%$ | $59.8 \pm 21.7\%$ |

CTX-LC-PFN conditions the predictions on two context signals: learner identity $z_L$ and a training-set performance token $z_T$. To assess each signal's individual contribution, we evaluate two ablations, CTX-LC-PFN (L) using only $z_L$ and CTX-LC-PFN (T) using only $z_T$, against the full CTX-LC-PFN (L+T) model via per-*(dataset, learner)* majority vote win rate on each extrapolation point NLL, averaged over five seeds (Table 1). Learner identity is the dominant signal, since CTX-LC-PFN wins only around 50%–55% of pairs against CTX-LC-PFN (L), statistically indistinguishable from a tie, indicating that $z_L$ alone already captures most of the contextual gain. Although $z_L$ alone accounts for the bulk of the gain, we use two tokens $z_L+z_T$ as the default configuration because the performance token is free to obtain and conditioning on both can reach a lower training loss.

# C. Interpretability of Learner Token Geometry

To understand what the learner tokens capture, we visualize pairwise cosine similarities between their learned embeddings. Since the learner token is the only context component that varies with the learning algorithm, stable structure in this space indicates that CTX-LC-PFN encodes learner-specific learning behavior rather than acting as a content-free dummy token. The observed clusters therefore suggest that learner conditioning provides information beyond the stabilization effect of the dummy token ablation.

Figure 8 shows the mean cosine similarity across five random seeds and its standard deviation. The mean matrix highlights consistently close learners, while the standard deviation matrix shows stability across runs.

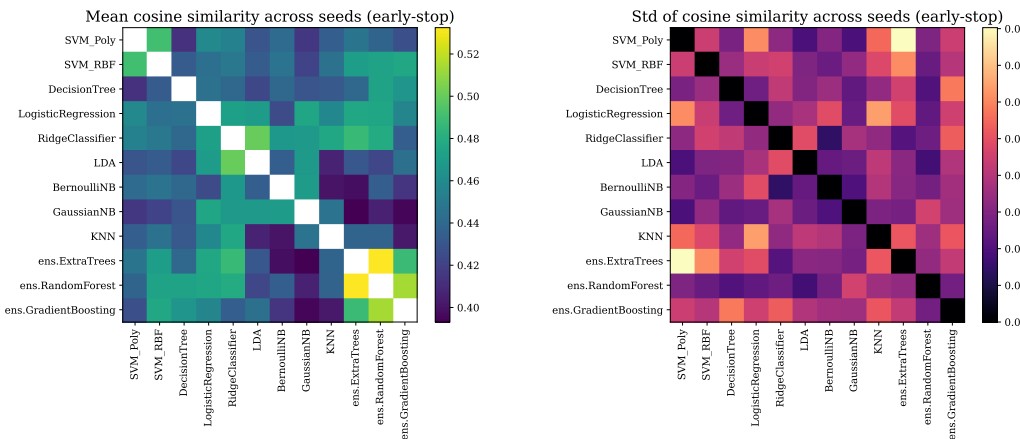

*Figure 8.* Mean pairwise cosine similarity between learner tokens across random seeds, together with the corresponding standard deviation.

Figure 9 repeats the clustering for each random seed. The exact merge order and some local groupings vary with the seed, but several broader relationships recur, suggesting that learner tokens capture meaningful coarse similarities even if the finer hierarchy is not fully stable.

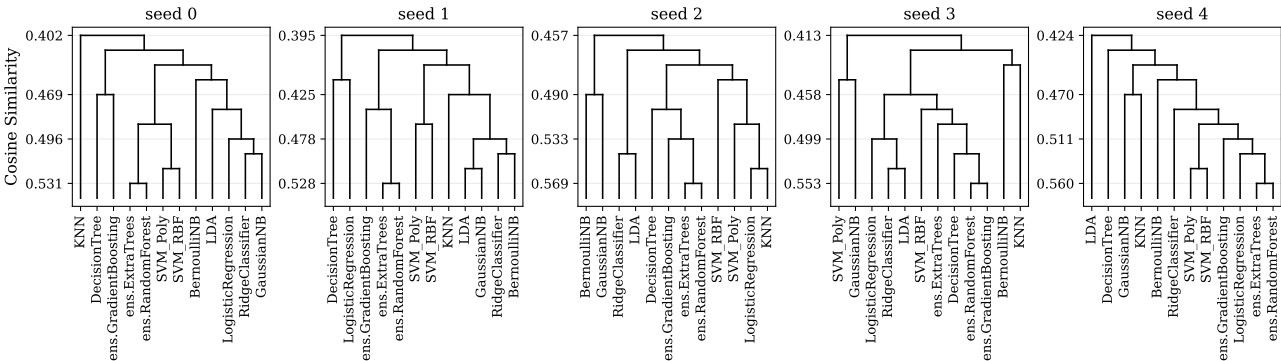

*Figure 9.* Per-seed average-linkage clustering of learner tokens, used to assess the consistency of the learned grouping structure across random seeds.

## D. Learning Curves Database 1.1 Clean

The largest tabular data learning curve benchmark, the Learning Curves Database 1.1 (LCDB 1.1) (Yan et al., 2025), contains learning curves between multiple learners in different tabular datasets. While LCDB 1.1 is challenging to extrapolate via parametric models and Transformer-based approaches (Cazacu et al., 2025), some curves are less informative due to issues on either the algorithm or the dataset side. On the algorithm side, we exclude some unstable iterative algorithms whose learning curves may fluctuate because training does not necessarily converge to an optimum under the default hyperparameter setting. We also remove algorithms that do not support all datasets (resulting in missing learning curves). On the dataset side, following common filtering practice (Bischl et al., 2021; Erickson et al., 2025), we remove datasets that are too small or too large, extremely imbalanced, contain large raw images, or are otherwise problematic. Figure 10 shows the distribution of dataset features and sample sizes in LCDB 1.1 before and after curation.

In experimental setup (Section 4), we split LCDB 1.1 Clean at the dataset level into 80/10/10 training, validation, and held-out test set for different random seed. All learners within a dataset remain in the same split, so no dataset contributes curves to more than one split. The model is trained only on curves from training set. Proxy validation loss and early stopping use validation set, which are unseen during training. The held-out test datasets are used for benchmarking after models are fixed. The procedure is repeated over five random seeds, and reported metrics are averaged across seeds.

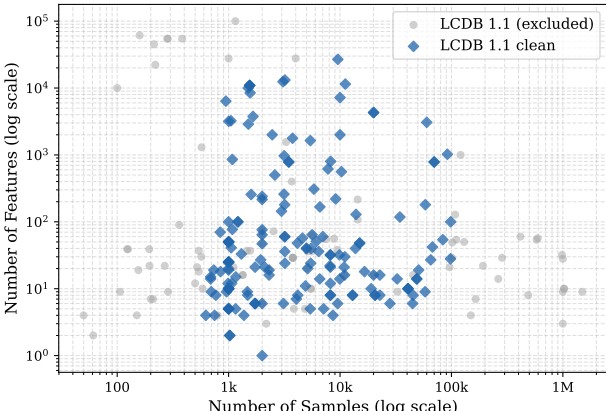

*Figure 10.* Distribution of dataset sizes (number of samples and features) in LCDB 1.1. Grey circles indicate datasets excluded by the curation criteria; blue diamonds indicate the retained LCDB 1.1 Clean subset.

# E. Evaluation Metrics

This appendix specifies the metrics used in Figure 3. Let $M$ denote the number of evaluated future prediction points. For each evaluated point $i \in \{1, \ldots, M\}$, let $y_i$ denote the held-out generalization-performance value at training size $x_i$, let $F_i$ denote the predictive cumulative distribution function, and let $\hat{y}_i$ denote the point prediction. For PFN-based models, $\hat{y}_i$ is the predictive median; for deterministic baselines, it is the fitted extrapolated value.

### E.1. Probabilistic Predicton Metrics

**Negative log-likelihood (NLL).** NLL evaluates whether the predictive distribution assigns high probability density to the observed future generalization-performance values. We compute

$$\text{NLL} = -\frac{1}{M} \sum_{i=1}^{M} \log q_i(y_i), \tag{1}$$

where $q_i(y_i)$ denotes the density induced by the predicted bin mass and corresponding bin width at the observed value. Lower NLL indicates higher density assigned to the observed future generalization-performance values. For deterministic baselines, $q_i$ is obtained from the calibrated binned distributions described in Appendix A.

**Mean squared calibration error (MSCE).** MSCE measures quantile calibration (Kuleshov et al., 2018): a calibrated model should have approximately a fraction $p$ of observations below its predicted $p$-th quantile. For $m = 15$ confidence levels $\{p_j\}_{j=1}^m$, we compute

$$\text{MSCE} = \frac{1}{m} \sum_{j=1}^{m} (p_j - \hat{p}_j)^2, \tag{2}$$

where $\hat{p}_j = M^{-1} \sum_{i=1}^{M} \mathbb{I}\{y_i \leq F_i^{-1}(p_j)\}$ is the empirical coverage at level $p_j$. We compute MSCE directly on the held-out test datasets as a calibration diagnostic.

**Quantile loss.** Quantile loss, also known as pinball loss, evaluates the accuracy of predicted quantiles and penalizes under- and over-estimation asymmetrically (Koenker & Hallock, 2001). For a target quantile level $\tau \in (0, 1)$ and predicted quantile $q_\tau = F^{-1}(\tau)$, it is

$$\mathcal{L}_\tau(y, q_\tau) = \max\left\{\tau(y - q_\tau), (\tau - 1)(y - q_\tau)\right\}. \tag{3}$$

We report the average loss over 15 equally spaced quantile levels on the held-out test datasets.

### E.2. Point-Prediction Metrics

**Mean squared error (MSE).** MSE measures prediction error directly in performance space:

$$\text{MSE} = \frac{1}{M} \sum_{i=1}^{M} (y_i - \hat{y}_i)^2. \tag{4}$$

It emphasizes larger absolute errors and is reported on the generalization-performance scale of the learning curves.

**Root mean squared log-error (RMSLE).** RMSLE computes the error after a logarithmic transform:

$$\text{RMSLE} = \sqrt{\frac{1}{M} \sum_{i=1}^{M} \left(\log(1 + y_i) - \log(1 + \hat{y}_i)\right)^2}. \tag{5}$$

The log transform emphasizes relative deviations, which can make errors more comparable across low- and high-performing regimes (Alabdulmohsin et al., 2022).

### E.3. Data Estimation Error

As an additional proxy for data collection impact, we report a weighted point prediction error that penalizes mistakes at larger training sizes more strongly:

$$\mathcal{E}_{\text{data}} = \frac{1}{M} \sum_{i=1}^{M} |x_i(y_i - \hat{y}_i)|. \tag{6}$$

This metric follows the intuition that a comparable performance residual can imply more serious consequences for budget estimation at larger data scales. We therefore rank methods within each task rather than averaging $\mathcal{E}_{\text{data}}$ directly across tasks.

To motivate this proxy, assume a fixed unit cost per sample. Under a local approximation, the same performance residual $\Delta y_i = y_i - \hat{y}_i$ is more consequential at larger training sizes $x_i$, where mis-estimation corresponds to more wasted or missing samples.

**Proposition E.1.** *For a power-law learning curve $y(x) = ax^b + c$ under local log-linear approximation, the error in the estimated training set size $|\Delta x_i|$ at evaluation point $i$ satisfies:*

$$|\Delta x_i| \approx \frac{x_i}{|\beta|} |\Delta y_i|, \tag{7}$$

*where $\beta$ is the local slope parameter and $|\Delta y_i|$ is the performance prediction error.*

Let the learning curve be $y(x) = ax^b + c$. Around a reference training size $x_0$, define $z = \log x$ so that $x = e^z$ and $y(z) = ae^{bz} + c$. A first-order Taylor expansion around $z_0 = \log x_0$ gives

$$y(z) \approx y(z_0) + y'(z_0)(z - z_0). \tag{8}$$

With the local log-space slope $\beta := y'(z_0) = abe^{bz_0} = abx_0^b$ and intercept $\alpha := y(z_0) - \beta z_0$, this becomes the local log-linear approximation

$$y \approx \alpha + \beta \log x. \tag{9}$$

This approximation also gives $dy \approx \beta \, d\log x = \beta \, dx/x$: a small performance change corresponds to a relative change in data size, scaled by the local slope $\beta$. Rearranging yields

$$\frac{dx}{dy} \approx \frac{x}{\beta}. \tag{10}$$

Therefore, for an evaluation point $i$ with training size $x_i$ and performance residual $\Delta y_i = y_i - \hat{y}_i$, a first-order error propagation gives

$$|\Delta x_i| \approx \left| \frac{dx}{dy} \right| |\Delta y_i| \approx \frac{x_i}{|\beta|} |y_i - \hat{y}_i|. \tag{11}$$

# F. Statistical Significance Testing

**Rank Statistical Significance Testing.** For each evaluation cutoff, we perform pairwise comparisons between all curve models using stored per point negative log-likelihood (NLL) values. For each model pair, we collect paired raw NLL values from all evaluated prediction points across the five random splits. Each paired sample corresponds to the NLL assigned by the two models to the same prediction point under the same split and cutoff. We then apply a two-sided Wilcoxon signed-rank test (Wilcoxon, 1992). Since multiple model pairs are tested within each cutoff, we apply Holm's correction (Holm, 1979) to control the family-wise error rate at $\alpha = 0.05$. For visualization, we report a single grid of critical diagrams across cutoffs. Model positions are determined by their average rank computed from per point NLL values, where lower NLL receives a better rank. Red horizontal bars connect groups of models whose pairwise differences are not statistically significant after Holm correction. Since the number of evaluated prediction points depends on the cutoff, larger cutoffs contain more paired NLL samples.

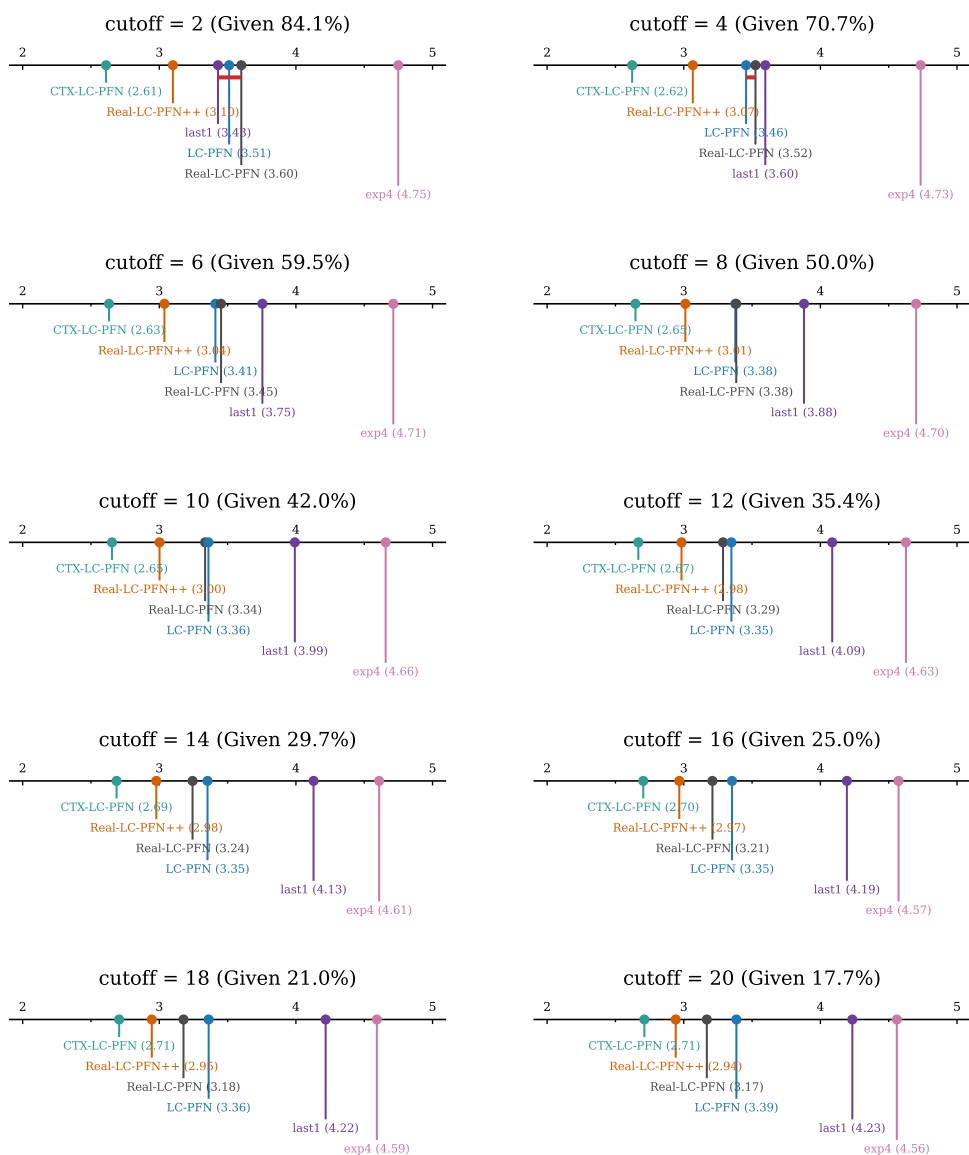

*Figure 11.* Critical diagrams for NLL across evaluation cutoffs. Model positions show average ranks computed from paired per point NLL values; lower ranks indicate better NLL. Red horizontal bars connect models whose pairwise NLL differences are not statistically significant after Holm correction at $\alpha = 0.05$.

**Per-Seed (Dataset, Learner)-Level NLL Comparison.** To complement the cutoff-wise critical diagrams, we provide a per-seed paired comparison of CTX-LC-PFN against Real-LC-PFN at the (dataset, learner) granularity. For each of the five dataset-level splits, we average each model's per point NLL over evaluation cutoffs and over the inner repetitions and prediction horizons within every (dataset, learner) pair, yielding $17 \times 12 = 204$ paired scalars per seed. We then apply a two-sided Wilcoxon signed-rank test to each seed independently; the reported $p$-value tests the null hypothesis that the median of the paired NLL differences is zero. Across all five seeds, CTX-LC-PFN attains lower mean NLL than Real-LC-PFN on the large majority of (dataset, learner) pairs, with $p \ll 10^{-4}$ in every seed (Figure 12).

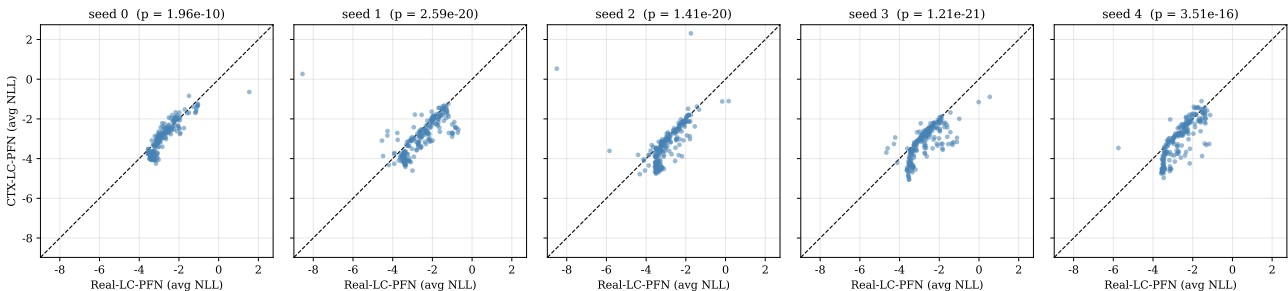

*Figure 12.* Per-seed scatter of (dataset, learner)-level mean NLL: CTX-LC-PFN ($y$-axis) versus Real-LC-PFN ($x$-axis). Each point is one (dataset, learner) pair (204 in total), with NLL averaged over cutoffs, inner repetitions, and prediction horizons for the corresponding seed. Points below the diagonal favor CTX-LC-PFN. Each panel reports the two-sided Wilcoxon signed-rank $p$-value over the 204 paired scalars for that seed.

