# OpenReview forum: "Context-Aware Learning Curve Extrapolation with Prior-Data Fitted Networks"
_ICML.cc/2026/Workshop/FMSD — FMSD @ ICML 2026 Poster_

### Official Review · Reviewer_UTE8 · 2026-05-20
**Review of Context-Aware Learning Curve Extrapolation with Prior-Data Fitted Networks**

**Rating:** 6
**Confidence:** 3

**Review:**

**Summary:**

The paper studies sample-wise learning curve extrapolation: given a partial learning curve showing model performance at smaller training set sizes, predict future performance at larger training sizes. The authors build on LC-PFN / Real-LC-PFN, which frame learning curve extrapolation as an in-context prediction problem using Prior-Data Fitted Networks. The paper first shows that directly training LC-PFN on real learning curves can overfit, and proposes Real-LC-PFN++, which stabilizes training using early stopping and a content-free dummy/register token. Building on this, the authors propose CTX-LC-PFN, which conditions the extrapolator on auxiliary context, specifically learner identity and the last observed training-set performance. On LCDB 1.1 Clean, a tabular learning curve benchmark, CTX-LC-PFN improves probabilistic extrapolation and calibration metrics such as NLL, MSCE, and quantile loss, with more modest but generally positive improvements on point-prediction metrics.

**Strengths:**

1. **Well-motivated problem:** Learning curve extrapolation is practically useful for estimating data requirements, deciding whether additional data collection is worthwhile, and predicting future model performance.
2. **Simple and effective method:** Conditioning on learner identity and training performance is intuitive. Different learners can have very different learning curve shapes, and training performance can provide useful information about task difficulty.
3. **Good empirical setup:** The evaluation uses LCDB 1.1 Clean, dataset-level splits, multiple observation regimes, multiple metrics, and five random seeds. Along with the fact that it borrows a similar experimental setup to the baseline methods it evaluates, this gives the empirical results reasonable credibility. Further, the paper evaluates not only point prediction but also uncertainty and calibration through NLL, MSCE, and quantile loss. This is important because learning curve extrapolation should ideally provide calibrated uncertainty, not just a single future estimate.
4. **Good stabilization analysis:** The diagnosis of Real-LC-PFN overfitting and the Real-LC-PFN++ ablation are useful. The dummy token and early stopping appear to provide a stronger real-data baseline before adding informative context.
5. **Good workshop fit:** The paper is aligned with the workshop theme. It studies tabular-data learning curves to determine the amount of data required to train a performant model.

**Areas for Improvement:**
1. **Contribution is incremental:** The main method is essentially adding context tokens to Real-LC-PFN, plus stabilizing real-data training with early stopping and a dummy token. This is sensible and useful, but the methodological novelty is limited.
2. **Clarify tabular-specific motivation:** The paper cites prior work showing that tabular learning curves are heterogeneous and sometimes non-monotone, but it could more explicitly explain what makes tabular learning curve extrapolation different from other domains. This is a minor issue, since the prior LC-PFN / Real-LC-PFN literature already motivates this setting, but a short clarification would help readers outside this subarea.
3. **Clarify whether the method is tabular-specific or domain-general:** Architecturally, CTX-LC-PFN appears to be a general learning curve extrapolator: it consumes partial curves and context tokens, not raw tabular rows. The tabular specificity comes from the training distribution and evaluation on LCDB 1.1. The paper should clarify whether the method is intended as a tabular-specific extrapolator or as a general learning curve extrapolation method currently trained on tabular curve corpora.
4. **Learner identity assumes known and seen learners:** The method conditions on learner identity, but the current setup keeps learner identities known at test time. It would be useful to evaluate whether the method generalizes to unseen learners, or to represent learners using descriptors or hyperparameters rather than only learned ID tokens.
5. **Context design space is limited:** The paper uses learner identity and observed training performance, but does not explore richer context such as dataset meta-features, number of classes, number of features, imbalance, learner hyperparameters, or simple curve-shape statistics. Since context conditioning is the central contribution, broader context ablations would strengthen the paper.

**Justification of Score:**

Overall, this is a simple, solid, and relevant workshop paper. The problem is important, the method is simple and well motivated, and the experimental setup is appropriate. The paper’s strongest contribution is showing that real-data LC-PFN training can be stabilized and that learner/context conditioning improves probabilistic extrapolation and calibration on LCDB 1.1. The main limitations are that the method is incremental, the context design space is only lightly explored, and the paper could better clarify whether the proposed approach is tabular-specific or broadly applicable beyond tabular learning curves. These issues do not undermine the main empirical findings, but they limit the strength and generality of the contribution.

---

### Official Review · Reviewer_XHmQ · 2026-05-21

**Rating:** 7
**Confidence:** 3

**Review:**

### Summary
The paper addresses the question of sample-wise learning curve extrapolation for tabular ML. The authors observe that Real-LC-PFN can overfit and introduce Real-LC-PFN++, which basically adds validation loss based early stopping and register token to stabilize the training. On top of this, they propose CTX-LC-PFN, which basically adds auxilary context such as learner identity and the last observed training set performance. Overall, evaluation is solid and shows that Real-LC-PFN++ and CTX-LC-PFN give improved NLL, quantile loss and calibration (see figure 3).

### Strengths
1. The paper demonstrates that Real-LC-PFN suffers from overfitting, and the early stopping based solution together with register token based solution is quite simple yet effective to mitigate this issue. To my understanding, this is the main source of performance improvement in this paper.  On top of this, for CTX-LC-PFN context tokens are prepended to the observed curve, and the PFN predicts future values in one forward pass, which is again a simple but effective solution.
2. The related work is quite comprehensive, well written and situates the paper well into the literature. Similarly, the paper is very well written with good exposition.
3. Evaluation is broad. The authors report probabilistic metrics, point prediction metrics, and includes comparisons to LC-PFN, Real-LC-PFN, Real-LC-PFN++. They also repeat experiments over five random dataset-level splits.

### Areas for Improvement
1. The contribution is somewhat incremental. CTX-LC-PFN mainly adds context tokens to an existing PFN framework and Real-LC-PFN++ has two interventions. This is not to say that the proposed methodology of the paper is not interesting/or valuable but it would be good to think how to extend the contributions of this paper. Maybe more deeply analyzing when and why each context source helps could be one way of doing it.
2. CTX-LC-PFN  setting assumes that learner identities are known at test time and remain within the same set of learners. This is limiting for the general context-aware extrapolation, for example, if we do encounter a new learner.
3. CTX-LC-PFN improvements are minor (but consistent) compared to Real-LC-PFN++? Is this because in fact contextual knowledge does not help substantially or the existing context provided is not enough to give more significant benefits,

### Detailed Comments
I liked reading this paper and thank you for submitting such a well-written manuscript. I think adding a few contributions beyond the existing ones would be important if you want to submit this work to an archival venue. I think adding an unseen learner experiments in line with my comment #2 would also be interesting to the audience.

---

### Official Review · Reviewer_ecP1 · 2026-05-21

**Rating:** 6
**Confidence:** 3

**Review:**

## Summary
This paper tackles the challenge of sample-wise learning curve extrapolation using Prior-Data Fitted Networks (PFNs). They found that directly training Real-LC-PFN on real-world data often leads to overfitting and propose two improvements. First, they introduce Real-LC-PFN++, a stabilized baseline that employs early stopping based on validation loss and integrates a learnable dummy token. Second, they introduce CTX-LC-PFN, which replaces the dummy token with two task-specific tokens: (1) the learner identity that encodes which classifier produced the curve, and (2) the latest observed training-set performance. Evaluated on the LCDB 1.1 Clean benchmark, CTX-LC-PFN demonstrates the superior performance.

## Strengths
 - I find the proposed methodology (register tokens, either dummy or based on the learner identity and training-set accuracy) intuitive and well-motivated.
 - Experiments include several metrics, the use of five random dataset-level splits and statistical significance testing. Several ablation studies are provided.
- The analysis of the learner identity tokens using average-linkage hierarchical clustering is a good addition.

## Areas for Improvement
 - The model relies on a one-hot indicator and a learned embedding lookup to encode the learner identity. Because the learner identities must be present in the training set and known at test time, the architecture is structurally restricted to the specific algorithms it was trained on. This prevents the model from being used zero-shot on novel, out-of-domain classifiers, which limits its utility.
- The non-PFN baselines are limited to a simple heuristic (last1) and a single parametric model (exp4). Given the sequential nature of the data, I wonder why not consider this task as a sort of time-series forecasting? Then, the  empirical evaluation could be significantly strengthened by including standard time-series forecasting baselines: simple methods like linear regression, or modern zero-shot time-series forecasting foundation models.

##  Detailed Comments
 - Although encoding the last observed training-set performance as a proxy for task difficulty is intuitive and effective design choice, I wonder why the authors restricted themselves to a single value? Did they try encoding the entire observed prefix of the training accuracy curve instead? This could provide the model with a much richer signal about the training dynamics (e.g., convergence speed and stability).
 - Since the learner identity uses a fixed lookup table, what is the authors' recommended fallback for a user who wants to evaluate a custom algorithm? The authors should discuss this limitation explicitly. A potential solution for future work could be to embed algorithms based on continuous meta-features (e.g., hyperparameter configurations or architectural traits) rather than a rigid categorical ID.
- Why the authors fix their methodology to the PFN architectures only? I think it could be interesting to try their ideas for other architectures. For example, one direction could be to fine-tune a time series foundation model.
- While the experimental setup is designed to prevent a data leakage, I wonder if the authors studied the robustness of the approach in the wild. For example, when test datasets come from another domain, or when sample-wise learning curves are just manually constructed. This could benefit the analysis of the approach.

## Justification:
Overall, this is a good paper with sound methodology and a working solution. There are some things that could be improved, but the paper meets the bar for acceptance.